# Amoebiasis: Advances in Diagnosis, Treatment, Immunology Features and the Interaction with the Intestinal Ecosystem

**DOI:** 10.3390/ijms241411755

**Published:** 2023-07-21

**Authors:** Patricia Morán, Angélica Serrano-Vázquez, Liliana Rojas-Velázquez, Enrique González, Horacio Pérez-Juárez, Eric G. Hernández, Maria de los Angeles Padilla, Martha E. Zaragoza, Tobías Portillo-Bobadilla, Manuel Ramiro, Cecilia Ximénez

**Affiliations:** 1Laboratorio de Inmunología, Unidad de Investigación en Medicina Experimental, Facultad de Medicina, Universidad Nacional Autónoma de México (UNAM), Ciudad de México 06726, Mexico; 2Unidad de Bioinformática, Bioestadística y Biología Computacional, Red de Apoyo a la Investigación, Coordinación de la Investigación Científica, Universidad Nacional Autónoma de México (UNAM)-Instituto Nacional de Ciencias Médicas y Nutrición Salvador Zubirán, Ciudad de México 14080, Mexico; 3División de Estudios de Posgrado, Facultad de Medicina, Universidad Nacional Autónoma de México (UNAM), Ciudad de México 04510, Mexico

**Keywords:** amoebiasis, advanced diagnosis and treatment, pathogenesis, intestinal microbiota interaction

## Abstract

This review of human amoebiasis is based on the most current knowledge of pathogenesis, diagnosis, treatment, and Entamoeba/microbiota interactions. The most relevant findings during this last decade about the *Entamoeba* parasite and the disease are related to the possibility of culturing trophozoites of different isolates from infected individuals that allowed the characterization of the multiple pathogenic mechanisms of the parasite and the understanding of the host–parasite relationship in the human. Second, the considerable advances in molecular biology and genetics help us to analyze the genome of *Entamoeba*, their genetic diversity, and the association of specific genotypes with the different amoebic forms of human amoebiasis. Based on this knowledge, culture and/or molecular diagnostic strategies are now available to determine the *Entamoeba* species and genotype responsible for invasive intestinal or extraintestinal amoebiasis cases. Likewise, the extensive knowledge of the immune response in amoebiasis with the appearance of new technologies made it possible to design diagnostic tools now available worldwide. Finally, the understanding of the interaction between the *Entamoeba* species and the intestinal microbiota aids the understanding of the ecology of this parasite in the human environment. These relevant findings will be discussed in this review.

## 1. Introduction

The genus *Entamoeba* consists of at least seven different species (*E. histolytica*, *E. coli*, *E. hartmanni*, *E. polecki*, *E. dispar*, *E. moshkovskii*, and *E. bangladeshi*) that can inhabit the human intestine and one (*E. gingivalis*) that can be found in the human oral cavity [1]. Until today, the only species recognized as an etiologic agent of the amoebic invasive disease is *Entamoeba histolytica*, with the large intestine as the main target organ [2].

Amoebiasis is defined as the gastrointestinal infection caused by *E. histolytica* with or without clinical manifestations [2], and is responsible for the worldwide morbidity and mortality rates due to amoebiasis in children and young adults. The global significance of amoebiasis is widespread, with the highest burden of amoebiasis borne by those residing in developing countries, particularly the tropics and subtropics, where there are inadequate hygiene levels and access to sanitation [3]. The efforts being made in the study of amoebiasis epidemiology in developing countries allow us to have a deeper understanding of its global panorama [4,5]. The estimations of the worldwide burden of amoebiasis by the World Health Organization (WHO) indicated that approximately 500 million people were infected with the parasite and 10% of these individuals had invasive amoebiasis [6]. Some reports mention that more than 55,000 people die each year due to acute intestinal amoebiasis [7,8].

Amoebiasis is caused by the ingestion of food or water contaminated with cysts, the infective form of *E. histolytica*. Following excystation, the emerging trophozoites migrate to the large intestine [9]. The trophozoites (mainly for *E. histolytica*) can become virulent and invasive, where they start to destroy the mucous–epithelial barrier, thus inducing the overproduction of mucus, killing host cells, and provoking inflammation, subsequently causing amoebic colitis [10]. *Entamoeba histolytica* can be transmitted by asymptomatic hosts infected with cysts and patients with intestinal amoebiasis, who are the natural transmitters [11]. The *Entamoeba* cysts are resistant to desiccation in soil and can survive in humid environments and food or water for several weeks, and susceptible hosts exposed to the infection sources ingest the cysts, which then undergo excystation during their pass through the gastrointestinal tract [11,12]. When in excystation, each cyst produces eight vegetative forms or trophozoites, which are the motile form of the parasite [11]. Trophozoites multiply by binary fission and some of them may encyst and be excreted with stools [13]. In asymptomatic infections, *Entamoeba* species trophozoites live as commensal feeding on colonic microbiota and nutrients of the host and forming cysts that pass through stools to perpetuate a new life cycle by fecal–oral spreading (Figure 1) [4]. It should be noted that more than 90% of infections have an asymptomatic course and are often self-limited over different periods [14].

Amoebiasis is endemic in the developing countries of Central and South America, Africa, and Asia [3,15]. However, until now, the exact burden of amoebiasis in these countries has been difficult to determine, and reports can be affected by geographic region, study design, sample size, incubation, symptom severity, and the sensitivity of the diagnostic modality used. In addition, diagnostic capacities and surveillance are often limited in areas where *E. histolytica* is endemic [4,16].

## 2. Clinical Features

Following the 1997 documents by the World Health Organization, amoebiasis is defined as the infection of the protozoan parasite *E. histolytica* with or without clinical manifestations [2]. The spectrum of the disease ranges from asymptomatic infection to the development of a severe infection with amoebic colitis and/or amoebic liver abscesses [17]. Chronic non-dysenteric colitis is the most frequent form of amoebiasis in people of all ages, characterized by non-specific symptoms [3]. After exposure to the parasite, 80% of patients display symptoms over a few days to 4–6 weeks [11]. The natural history of invasive intestinal amoebiasis is an acute event, characterized by the presence of diarrhea that occurs days or weeks after exposure and does not last more than four to five weeks, although there are reports of its occurrence years later after exposure to a source of infection [11]. In the latter case, it is assumed that the cause–effect relationship is extremely difficult to corroborate [18]. The natural history of this clinical form can also mimic irritable bowel syndrome. Symptoms more relevant in this instance are periods of abdominal pain (colic) and auto-limited episodes of diarrhea alternating with constipation. However, both non-dysenteric amoebic colitis and irritable bowel syndrome are controversial themes in clinical practice [11]. A few individuals with *E. histolytica* infections develop extraintestinal features including liver abscesses, purulent pericarditis, lung abscesses, pneumonia, peritonitis skin lesions, and even cerebral abscesses, due to the dissemination of trophozoites via the blood circulatory system [19,20,21]. Of these clinical features, the liver abscess is the most common manifestation, and delay in diagnosis and treatment may cause fatalities.

### 2.1. Amoebic Liver Abscess

Amoebic liver abscess is a disease that can affect individuals of all ages; in some endemic areas, the incidence rates are higher in both children under 5 years and young adults. Males are also more prone to developing amoebic hepatic abscesses than females [3,11]. The most common symptoms suggestive of amoebic liver abscess are fever (38 °C), chills and diaphoresis, anorexia, and abdominal pain in the right upper quadrant that increases during inspiration. Pain also frequently radiates to the shoulder and back. Nausea has also been referenced but diarrhea is only occasionally mentioned (50% of cases) [11,22].

Hepatomegaly can be detected during digital percussion of the hepatic area and is always related to the dimensions of the abscess; patients can also display peritoneal signs (abdominal guarding or rebound). The absence of intestinal noises, jaundice, and pleural or pericardial rub are symptoms that should elicit alarm related to the rupture or imminence of rupture of the hepatic abscess [23].

### 2.2. The Severe Forms of Amoebiasis

The severe forms of amoebiasis are colon ameboma, fulminant necrotizing colitis, and toxic megacolon [24]. The appearance of symptoms, such as severe dysentery and pain with signs of peritoneal irritation (rebound), intense tenesmus, a fever (>38 °C), tachycardia, hypertension, nausea, and anorexia, are suggestive of the previously mentioned severe forms of intestinal amoebiasis [20]. The mortality rates of dysenteric syndrome due to *E. histolytica* are less than 1%, but mortality due to complications increases up to 75% [3]. Fulminant amoebic colitis is a rare complication of amoebiasis associated with high mortality, and can occur in more than 50% of cases with severe colitis [20,24]. The severe forms of invasive amoebiasis can be observed in young children, pregnant women, the elderly, and those with chronic diseases and individuals being treated with immunosuppressants or those with immunodeficiency disorders [11].

The circumstances that define the extraintestinal invasive behavior of some *E. histolytica* strains remain unknown today. However, this seems to be an infrequent event as suggested by the low morbidity rates of amoebic liver abscess and other extraintestinal forms of invasive amoebiasis compared with the prevalence rates of asymptomatic infections and intestinal disease, and fortunately, in the last decade, such complications are uncommon [25,26].

## 3. Diagnosis

Amoebiasis is an acute disease in which the most frequent symptoms are abdominal pain (colic) and the presence of diarrhea with mucus and/or blood, or a clear dysenteric syndrome, but fever and other systemic symptoms are infrequent [27]. In endemic areas, patients with clinical signs and symptoms including gastrointestinal upset and watery or bloody diarrhea should be suspected of intestinal amoebiasis and should be treated [28].

### 3.1. Macroscopy and Microscopy Analyses

The gold standard for the diagnosis of amoebiasis is the identification of the parasite, mainly in the feces of people infected with *Entamoeba*, where cysts and, rarely, trophozoites are identified. This identification is carried out by direct microscopy examination; the Faust–Ferreira concentration method has been the most widely used for microscopic identification [29]. While technically simple in approach, these techniques require the expertise of highly qualified technicians in the morphological identification of ova, cysts, and trophozoites to be feasible, and the sensitivity and specificity are no more than 80% [30]. This technique cannot differentiate between *E. histolytica* or *E. dispar*, but in some endemic communities, this is the only diagnostic technology available [31]. Where endoscopy examination is available, a colonoscopy can be of great help in the clinical diagnosis of invasive intestinal amoebiasis [20]. This procedure allows for the microscopic examination of samples taken directly from the characteristic flask-shaped ulcer produced by *E. histolytica* and from other sites of mucosal lesions.

On the other hand, a colonoscopy detects the presence of lesions related to the mentioned severe forms of intestinal amoebiasis and allows for the differential diagnosis of other pathologies, such as inflammatory bowel disease or colon carcinoma (Figure 2) [11].

### 3.2. Imagen Diagnosis

In amoebic liver abscesses, in general, the right hepatic lobule is the most frequently affected due to the portal circulatory system of the right colon; however, the left lobule can be also affected [20]. Laboratory findings suggestive of amoebic liver abscess are the presence of leukocytosis, neutrophilia, increased globular sedimentation velocity, and high levels of alkaline phosphatase [3,32].

Thoracic X-ray data useful in the diagnosis of amoebic abscesses, as well as other types of hepatic abscesses, include the elevation of the right hemidiaphragm, atelectasis, or pleural effusion [3,33]. An ultrasound is the gold standard technique for the diagnosis of amoebic liver abscesses and is of tremendous importance in countries with limited medical and economic resources [34]. Several authors have mentioned the presence of a large single abscess as a frequent characteristic of amoebic liver abscesses; however, this characteristic is not a determinant of amoebic abscesses, and the pyogenic abscesses are also not characteristic of multiple abscesses [35,36].

Early lesions due to the amoebic invasion of hepatic parenchyma are multifocal (micro-abscesses) because of tissue destruction and necrosis by proteases from *E. histolytica*, and neutrophil recruitment to the site of infection [37]. The advantage of computed tomography (CT) scans and magnetic resonance is the detection of small abscesses and the high definition of the images. Moreover, other techniques not always available in endemic countries (e.g., gallium scans) can help differentiate between amoebic (cold images) and pyogenic abscesses (hot images). Thus, the difference is based on the absence (amoebic) or presence (pyogenic) of white blood cells in the abscesses [36,38].

### 3.3. Immunological and Molecular Diagnosis

During the last 10 years, diagnostics in amoebiasis have changed dramatically, considerably improving sensitivity and specificity. Some of the current techniques are based on immunological strategies, such as ELISA and different modalities of polymerase chain reaction (PCR). There are clear advantages in the bedside diagnosis and clinical laboratories of health institutions [31].

In our experience, immunologically based diagnostics tests for the detection of antiamoebic antibodies in serum are excellent tests in hospitals, particularly in patients with liver abscesses, and their results are reproducible and reliable to perform the differential diagnosis between amoebic and pyogenic liver abscesses and determine the appropriate treatment [39,40]. The presence of the parasite can also be identified in the drainage of liver abscesses, even though trophozoites are not detected in these types of samples, it is possible to obtain the DNA of the parasite through molecular techniques. Likewise, in tissue biopsies of cutaneous or intestinal amoebiasis lesions, microscopic identification of the trophozoites and/or by PCR is possible. Advances in molecular diagnostics have improved our understanding and led to the recognition and separation of *E. histolytica* from nonpathogenic species of *Entamoeba*. Therefore, PCR has become a very sensitive and specific diagnostic method [31,40].

ELISA tests have been used both for the detection of antiamoebic antibodies in serum and for the capture of amoebic antigens in feces. However, the difficulty of differentiating a past infection from an active one considerably limited its use to identify patients requiring treatment; instead, it has been used for seroepidemiological studies [41,42,43], so there are many commercial tests that can even differentiate *E. histolytica* from *E. dispar*. These diagnostic kits allow for the generation of reliable results quickly by following simple instructions, but they are expensive due to the use of monoclonal Abs (mAbs) for antigen capture and are not routinely used in clinical laboratories.

Of these kits, the most popular mAbs for use against Gal/Lectin GalNAc are: the *E. histolytica* TechLab kit and the CELISA PATH kit (Cellabs, Brookvale, Australia), which were developed specifically to detect *E. histolytica* in stool samples, and the TechLab *E. histolytica* II (Blacksburg, VA, USA), which allows the detection of different epitopes between the Gal-lectins of *E. histolytica* and *E. dispar* [44,45,46]. The sensitivity and specificity of the TechLab kits I and II vary by 80–99% and 86–98%, respectively [44]. Other kits include the Optimum S kit (Merlin Diagnostika, Bornheim-Hersel, Germany), which detects the components of the family of serine-rich proteins, the Alexon ProSpecT ELISA (Remel Inc., Lenexa, KS, USA), which detects an “AND *E. histolytica*-specific antigen” (EHSA) from *E. histolytica*/*E. dispar*, and the Triage Parasite Panel (BIOSITE Diagnostics, San Diego, CA, USA), which detects a 29-kDa antigen from *E. histolytica*/*E. dispar* using an immune-chromatographic strip (118). Recently, an ELISA sandwich assay was developed using two mAbs raised against a spacer region of the Jacob2 lectin of *E. histolytica*, a protein of the cyst wall [47]. Notably, the mAbs only recognized the cysts of *E. histolytica* but did not recognize the cysts of *E. dispar* or *E. Bangladeshi* by immunofluorescence assays. The successful detection of cysts in formalin-fixed stool samples was observed, suggesting that the anti-Jacob2 mAbs could be used for the development of a diagnostic kit for *E. histolytica* infection in any type of stool sample.

Recently, several molecular diagnostic methods, including conventional and real-time polymerase chain reactions (PCRs), have been developed for the specific diagnosis of each of these three *Entamoeba* species in clinical laboratories. Currently, there is a great variety of PCR methods targeting different genes for the detection and differentiation of these amoebas [31,48]. The consistent genetic diversity detected in the small rRNA subunit (18SDNAr) of *E. histolytica* and *E. dispar* initiated the use of this gene as a target for the differentiation of the two species [49,50,51,52,53]. PCR techniques targeting this gene are 100 times more sensitive than the current best available enzyme immunoassay (ELISA) kit [54,55,56] and are widely used to distinguish these species since there are multiple copies of these genes in amoebas [57], which makes the 18SDNAr gene easier to detect than a DNA fragment from a single copy of a gene.

However, for a good result from the PCR, good-quality DNA is required, which is why extraction optimization is necessary. Recently, simple and effective methods for the isolation of *Entamoeba* DNA from feces have been developed. Commercially available DNA extraction kits are recommended due to their ease and success as they minimize extraction time, and the DNA can be extracted directly from the feces without the need to culture the parasite. The following kits have been used for the direct extraction of DNA from fecal samples include XTRAX DNA extraction kit [58] (Gull Laboratories, Salt Lake City, UT, USA), Extract Master Faecal DNA extraction kit Epicentre Biotechnologies, Madison, WI, USA), and the Genomic DNA Prep Plus kit [59]. Of the methods for DNA extraction, Qiagen kits have been used the most widely with the QIAamp stool kit now predominating. The QIAamp tissue kit spin columns (QIAGEN) have also been used for the extraction and purification of *Entamoeba* DNA from fecal samples [60]. Various modifications, including optimizing the duration and temperature of proteinase K digestion, adding additional wash steps, and treatment of the sample with 2% polyvinylpolypyrrolidone, have all been used [61,62]. The QIAamp stool kit has proven to be the most widely accepted method for DNA extraction providing successful and reliable recovery of DNA from fecal material [50,63,64]. Aspirated pus from ALA can also be tested by PCR and has shown excellent sensitivity and specificity [65,66].

## 4. Treatment and Prevention

Due to the different amoebiasis forms and severity of the symptoms, the treatment varies, and diverse drugs can be used. Regarding treatment, amoebiasis is usually treated by amebicides, which are prescribed depending on the severity of the infection [67].

Antiamoebics are the most recommended drugs; these are classified as (1) luminal amebicides which included paromomycin, diloxanide furoate, iodoquinol, and nitazoxanide; they act only on the intestinal lumen and are used to treat amoebic, not dysenteric, colitis, and (2) tissue amebicides such as chloroquine, emetine, tinidazole, and metronidazole, which are the elected treatment in patients with symptomatic intestinal amoebiasis, asymptomatic cyst carriers, and amoebic liver abscesses, due to their rapid intestinal absorption acting at the systemic level [15,68]. Metronidazole is the major drug used in invasive amoebiasis cases; although the toxicity and side effects of this compound are important, the most frequent side effects are nausea, vomiting, headaches, a metallic or bitter taste in the mouth, and some more serious effects such as anorexia, ataxia, and skin rashes/itching. Nevertheless, metronidazole is widely used in the treatment of invasive amoebiasis due to its efficiency and its low cost [69,70,71].

Likewise, the development of resistance has been reported in some anaerobic pathogens [72,73], but in our experience, there is no clear evidence of resistant strains of *E. histolytica* in patients with amoebic liver abscesses. However, it is important that after systemic treatment of invasive amoebiasis such as amoebic liver abscesses to prescribe a luminal antiamoebic treatment with the purpose of avoiding *E. histolytica*/*E. dispar* cysts luminal infection [15].

On the other hand, the primary measures for the prevention and control of amoebiasis infections remain hygiene habits, use of latrines, hand washing, improvement of water purification systems and adequate hygiene practices to prepare food, and the avoidance of fecal–oral exposure, which could all decrease disease incidence [74,75]. It seems easy to avoid amoebiasis infection by adopting these measures; however, these measures are unattainable when we talk about the fact that this infection is endemic in underdeveloped countries and in poor areas of these countries where the basic infrastructure is not available. Therefore, it is necessary that great efforts, on the part of governments and health systems, be directed to generate the conditions and infrastructure necessary to prevent and reduce the incidence of amoebiasis in developing countries.

Unfortunately, despite the efforts of the scientific community in the field, it has not been possible to obtain a vaccine against this disease so far. However, many recent vaccine development studies appear to be promising [20].

## 5. Pathogenesis

Pathogenesis of *E. histolytica* is multifactorial, since the virulent molecules of the parasite, as well as the host’s immune response, play an important role in the pathogenesis of the disease, causing damage to tissues that facilitate entry to systemic sites [76]. The destructive mechanisms of *E. histolytica* include contact with target cells, cytolysis, and phagocytosis, and the degradation of ingested cells. After contact with trophozoites in the epithelium, the cells increase the paracellular permeability produced by the opening of the TJ junctions and the distortion of the microvilli [77]. Superficial blisters and tiny focal discontinuities appear in the host plasma membrane, displacing and separating neighboring cells and compromising membrane integrity [78].

*Entamoeba histolytica* usually lodges in the intestine and in about 90 percent of cases, it generates an asymptomatic infection; however, it is not clear why a minority of individuals infected with *E. histolytica* activate a pathogenic phenotype [79]. Throughout many years of study, the participation of pathogenic molecules involved in various causal events due to *E. histolytica* infection has been demonstrated. The most studied physiological events are: (1) the colonization of the intestinal mucus layer, (2) the adherence of trophozoites to the host intestinal epithelium, (3) the interaction with the intracellular junctions of the epithelium, (4) cytolysis, (5) phagocytosis and trogocytosis, (6) the activation of the host’s immune response, and (7) the contributions of the host gut microbiota [4,78,80,81]

The genome sequence of *E. histolytica* strain HM1: IMSS was published and analyzed in 2005 [82,83,84]. The genome assembly contained 20, 800, and 560 bp of DNA in 1496 scaffolds. The genome had a high AT content (approximately 75%). Likewise, approximately half of the assembled sequence was predicted to be coding, with 8333 annotated genes [85]. Since it was sequenced in 2005, the genome of *E. histolytica* has been extensively analyzed due to its implications for human health [86].

These analyses of the *E. histolytica* genome have revealed a variety of metabolic adaptations, which include the reduction in or elimination of most mitochondrial metabolic pathways and the use of oxidative stress enzymes generally associated with anaerobic prokaryotes. Based on the knowledge of the *E. histolytica* genome there is evidence that suggests that cellular events such as meiotic recombination and the possible existence of a sexual reproduction mechanism have occurred in its chromosomes, which could allow the transfer of genetic information between different strains of *E. histolytica*, generating amoebas with a greater capacity to invade tissues and greater resistance to drugs, allowing them to survive in the host and even increase their capacity to survive outside it [87,88].

Another characteristic that has been observed in the *E. histolytica* genome is the so-called gain of genetic information, which includes the significant number of genes (at least 68) that seem to have been gained by horizontal transfer of bacterial genes from intestinal flora, some of them from the same donor group: alpha-1,2-mannosidase (EHI 009520), mannose-1-phosphate guanylyltransferase (EHI 052810), and fructokinase (EHI 054510) from Bacteroidetes; nicotinatephosphoribosyltransferase (EHI 023260) and hypothetical protein (EHI 072640) from Bacteroidales; and Fe-S cluster assembly protein NifU (EHI 049620) and metallo-beta-lactamase family protein (EHI 068560) from proteobacteria [82,84,89]. These genes tend to be involved in metabolic processes characteristic of the anaerobic lifestyle of the organisms [82]. Likewise, the analysis of the *E. histolytica* genome has also revealed that different genes encode proteins belonging to the A1G1 family, whose function is related to drug resistance in bacteria [90]. It has been observed that the expression of this group of proteins is strongly diminished in *E. dispar*; the previous study suggests that these molecules could participate directly in the virulence of the pathogenic amoebae [91]. Some genes that are important in the virulence of *E. histolytica* are genes that encode BspA-like proteins (BspA-like) that participate in cell aggregation by binding to extracellular matrix proteins and in adhesion to target cells; these functions are important for cellular damage and can also stimulate the production of proinflammatory cytokines through the stimulation of Toll-like receptors, which participate in the host innate immune response responsible for limiting tissue damage [92]. These cytokines are believed to help generate an inflammatory microenvironment that could favor the destruction of cells by amoebae [91]. The analysis of the *E. histolytica* genome has also made it possible to identify redundancy in genes that have been widely studied for their participation in different cellular processes related to the pathogenesis of *E. histolytica*. For example, the genes encoding the lectin Gal/GalNAc is important for the adhesion of trophozoites to epithelial cells; the genes that encode for proteins called amoebopores and some of the cysteine proteinases are involved in the cytolysis of epithelial cells [93]. Some of the molecules that have recently been identified, and the mechanisms involved in epigenetic modifications, such as DNA methylation or histone modification in *E. histolytica*, act directly in the conformation of chromatin and the regulation of gene expression [94]. These epigenetic regulatory mechanisms could be important to activate or prevent the expression of genes involved in virulence and cyst formation.

Another recent discovery in the study of the *E. histolytica* genome has been the presence of elements called transposons and retrotransposons, which constitute a significant portion of the genome of *E. histolytica* [95]. The insertion of transposons and retrotransposons results in changes in the amount of DNA, as well as changes and mutation genes. Although until now there is not much information about the function of this type of molecule, it is thought that these elements of amoebic DNA could be participating in the generation of pathogenic variants of *E. histolytica*, which are obtained from clinical isolates and cultured in the presence of pathogenic bacteria, in which there is an increase in the virulence of *E. histolytica*.

Another molecule that participates in *E. histolytica* infection is the cyclooxygenase from *E. histolytica* (EhCox). EhCox, derived from prostaglandin E2, stimulates the chemokine IL-8 from mucosal epithelial cells that recruit neutrophils to the site of infection, increasing the tissue damage. This phenomenon was verified by making gene silencing of EhCox (EhCoxgs) which increases the expression and production of endogenous cysteine protease (CP) and the virulence of trophozoites without altering CP gene transcripts. Shahi et al. propose a negative feedback mechanism in *E. histolytica* to limit proteolytic activity during colonization that can inadvertently trigger inflammation in the gut-enhanced erythrophagocytosis, cytopathic effects on colonic epithelial cells, and elicited proinflammatory cytokines in mice colonic loops, as well as increased inflammation associated with high levels of myeloperoxidase activity [76].

The sphingomyelinase enzymes encoded by *E. histolytica* are other molecules participating in the pathogenesis of three of the neutral sphingomyelinase enzymes encoded in the *E. histolytica* genome. In transfected trophozoites, an overexpression of EhnSM1 and EhnSM2 caused an increase in cytopathic activity and EhnSM3 induced an increase in hemolytic and cytotoxic activities. Another of the most important virulence molecules in *E. histolytica* is the amoebopore, which is a pore-forming protein. An increase in gene expression of amoebapores A, B, and C has been seen in virulent *E. histolytica* strains in comparison with less virulent *Entamoeba* species. Urquieta-Ramírez et al. [96] mention that in a *E. histolytica* strain that overexpresses neutral sphingomyelinases, the gene expression levels for cysteine proteinase 5, adhesin 112, and heavy and light Gal/GalNAc lectin subunits are not affected. They propose that the increase in the cytotoxic and lytic effect of EhnSM3 overexpressing strain can be related to the sum of the effect of EhnSM3 plus amoebapores in a process cell contact-dependent manner or as a mediator by inducing the gene expression of amoebapores enabling the link between EhnSM3 with the virulence phenotype of *E. histolytica*.

Efforts have also been made to elucidate the pathogenicity of *E. histolytica* from genomic studies. Wilson et al. [97] showed differences in the expansion and contraction of families of proteins associated with host or bacterial interactions and highlighted the importance to parasitic *Entamoeba* species of surface-bound proteins involved in adhesion to extracellular membranes, such as the Gal/GalNAc lectin and members of the BspA and Ariel1 families. On the other hand, Nakada-Tsukui et al. [98] find that the AIG1 family protein plays a pivotal role in *E. histolytica* virulence via the regulation of host cell adhesion.

The PTP superfamily is classified as classical tyrosine-specific phosphatases, dual-specificity phosphatases, cdc25 phosphatases, and low-molecular-weight protein tyrosine phosphatase (LMW-PTP) [99]. The genome of *E. histolytica* contains two genes codifying for LMW-PTPs, EhLMW-PTP1, and EhLMW-PTP2, which are identical to the amino acid sequence except for a single conservative residue change at position A85 V. Putative substrates have been identified using an EhLMW-PTP1 mutant at the catalytic site, suggesting it plays an essential role in parasite virulence [100]. Torres-Cifuentes et al. [101] analyzed the potential role of EhLMW-PTP2 in the virulence of *E. histolytica* and found a novel and unexpected effect on adhesion, phagocytosis, and cytopathic effect mediated by regulating the LMW-PTP2 mRNA and protein expression in transfected amoebas. They suggested that EhLMW-PTPs are essential enzymes that regulate cellular pathways activated in the parasite during infection.

On the other hand, it is known that iron is a determinant in the survival of *E. histolytica* and can modulate the expression of virulence factors. Some of the most representative events of the role of iron in the survival of *E. histolytica* are specified by Gastelum-Martínez et al. The most highly expressed virulence factors include EhCP1, EhCP2, EhCP3, and EhCP5. EhCP1-6 is overexpressed in limited iron conditions in culture. Negative regulation by iron was shown for Ehmmbp26 and Ehmmbp45, which obtain iron from the haem group [102]. Aldose reductase participates in siderophore biosynthesis and/or ROS formation. Superoxide dismutase (EhSOD) is responsible for the detoxification of superoxide radicals. Acetaldehyde/alcohol dehydrogenase-2 (EhADH2) is responsible for the internalization of human holo-transferrin. NADPH-dependent oxidoreductase (EhNO2) is involved in redox homeostasis through the reduction of L-cysteine and iron. Actin and genes related to the reorganization of the parasite cytoskeleton such as actobindin and cofilin are downregulated by iron [103]. The iron concentration could affect the cytoadherence capacity of the trophozoite. Acyl-CoA synthetase is involved in the catabolism of fatty acids. Cysteine synthase CS1 and CS2 and cysteine desulphurase (NifS) are involved in cysteine metabolism and possibly in the metabolism of sugar-containing amino acids, and both are upregulated by iron. S-adenosylmethionine synthetase (SAM) participates in the biosynthesis of SAM and many biosynthetic pathways including methionine-cysteine conversion and haem biosynthesis. SAM is downregulated by iron. The presence of cis-elements in mRNA (iron-responsive elements (IREs)) was demonstrated in the actin and Ehhmb26 genes, suggesting the presence of a post-transcriptional IRE/iron-responsive protein [IRP] iron regulatory system in this parasite.

During chronic inflammation, the vagus nerve limits the immune response through the anti-inflammatory reflex, which includes acetylcholine (ACh) as one of the predominant neurotransmitters at the infection site. Consequently, the response of *E. histolytica* trophozoites to ACh could be implicated in the establishment of invasive disease. Medina-Rosales et al. [104] evaluated the effect of ACh on *E. histolytica* virulence. They demonstrated that *E. histolytica* trophozoites bind ACh on their membrane and show a clear increase in the expression of virulence factors that were upregulated upon stimulation with the neurotransmitter. ACh treatment increased the expression of L220, Gal/GalNAc lectin heavy subunit (170 kDa), amoebapore C, cysteine proteinase 2 (ehcp-a2), and cysteine proteinase 5 (ehcp-a5). Moreover, erythrophagocytosis, cytotoxicity, and actin cytoskeleton remodeling were augmented after ACh treatment.

Lipids are essential in the cellular processes and pathogenesis of this organism *Entamoeba histolytica* has 27 putative enzymes involved in the lipid metabolic pathways and the trophozoites have mechanisms to synthesize or remodel phospholipids. They acquire cholesterol and fatty acids from the environment; thus, parasites need lipid scavenging from the host to regulate their cellular processes for survival or they may provoke disequilibrium in lipid synthesis and metabolism in the organism. However, lipid metabolism during host–pathogen interaction research is still an open field to be explored and exploited for the development of new methodologies for diagnosis and treatment [17].

## 6. Intestinal Ecosystem and *Entamoeba* Interactions

This is a very recent field of research and there are highly relevant findings on the interaction of *E. histolytica* in the colon with multiple components that modulate the invasion and destruction of tissues in the outer mucus layer [105]. In the colon, *E. histolytica* interacts with a diverse microbiota and takes advantage of its presence since it uses its decomposition products (such as glycans) that can serve as a source of nutrients, and at the same time, *E. histolytica* can feed on the colonic microbiota [105,106]. Recently, a metagenomics study on the in vitro association between *E. histolytica* and enteric bacteria showed that *E. histolytica* preferentially phagocytoses *Lactobacillus ruminus*, *Faecalibacterium prausnitzii*, *Bifidobacterium longum*, and *B. ruminantium*. These findings suggest that *E. histolytica* prefers to phagocytose commensal bacteria that are part of the healthy microbiota [106].

In addition to the predator–prey relationship, there is recent evidence that when the infection caused by *E. histolytica* is symptomatic, the intestinal microbiota could play an important role in the development of the disease [9,107]. Reyna-Fabian et al. examined samples aspirated from abscess material obtained from patients who were clinically diagnosed with amoebic liver abscesses (ALAs) or pyogenic liver abscesses (PLAs). They suggested that most of the ALAs or PLAs are mixed abscesses (bacteria and protozoa) and that the disease outcomes could be related to the bacterial/protozoan density ratio [35]. This is in line with Ximénez et al., 2010, who hypothesized that the size of intestinal amebic ulcers and the local inflammatory response permits the exit of *E. histolytica* trophozoites and intestinal bacteria, some of which enter the phagocytic vacuoles of trophozoites [27].

It has been determined in experimental models, that both the pathogenicity and virulence of *E. histolytica* are affected by the presence or absence of enteric bacteria and this could help explain why only about 10% of individuals infected with *E. histolytica* develop intestinal amoebiasis [105,108,109,110]. Evaluating the pathogenicity of *E. histolytica* with co-infection with other parasites has been reported in in vitro studies to show a twofold increase in CaCo monolayer destruction when *E. histolytica* interacted with enteropathogenic *Escherichia coli* (EPEC), but not with *E. coli* DH5α, for 2.5 h. This was associated with increased *E. histolytica* proteolytic activity as revealed by zymogram analysis and degradation of the *E. histolytica* CP-A1/5 (EhCP-A1/5) peptide substrate Z-Arg-Arg-pNC and EhCP4 substrate Z-Val-Val-Arg-AMC. Additionally, *E. histolytica*–EPEC interaction increased EhCP-A1, -A2, -A4, -A5, Hgl, Apa, and Cox-1 mRNA expression. In animal models of disease, the *E. histolytica*–EPEC interaction enhanced cellular inflammatory reaction, granuloma formation, and necrosis in ALA in hamsters and increased secretory and proinflammatory cytokine responses in closed colonic loops in mice [111].

Furthermore, Reyna-Fabian (2010) [35] showed that *E. dispar* in association with the gut microbiota could be potentially responsible for intestinal or liver tissue damage, like that observed in *E. histolytica*. More recently, Vilela da Silva et al. (2021) reported new evidence about some *E. dispar* strains in that the cysteine proteinase 5 expression in *E. dispar* MCR, VEJ, and ADO strains, isolated in Brazil, show that *S. typhimurium* and *E. dispar* co-infection worsens amoebic colitis, possibly by increasing the expression of amoebic virulence factors [112]. However, great effort is still required to elucidate the factors that drove these interactions relevant to the development of the disease caused by *E. dispar*.

On the other hand, important alterations in the intestinal microbiota composition have been observed due to infection by E. histolytica [9]. A dysbiotic state has been observed in linked E. histolytica-positive patients in India, characterized by a decrease in *Bacteroides*, *Clostridium coccoides*, *C. leptum*, *Lactobacillus*, *Campylobacter*, and *Eubacterium* with a corresponding increase in *Bifidobacterium* species [105,113]. Another recent study in Cameroon also linked the presence of *E. histolytica* with an increase in *Clostridial* and *Ruminococcaceae* and a corresponding decrease in *Prevotella copri* and *Fusobacteria* [114]. This study almost showed that in *E. histolytica*-infected individuals an augmented number of bacterial species (alpha diversity) and a decrease in interindividual variation (beta diversity) are evident. Also, it was recently demonstrated that a dysbiotic state renders the host hyper-responsive for increased proinflammatory cytokine and chemokine production with hypersecretory responses toward *E. histolytica* [105]. The authors mentioned that these findings are of great relevance, as individuals with dysbiosis (either by disease, antibiotics, or due to a poor diet) that are infected with *E. histolytica* are at a high risk of developing severe intestinal amoebiasis associated with acute inflammation, compared to individuals with a healthy microbiota. The fact that *E. histolytica* induces the production of antimicrobial peptides but is resistant to their cytopathic effect could explain the alteration in microbial composition observed in *E. histolytica* infections [115].

However, until now, it has been unknown whether the dysbiosis observed in amoebiasis is the cause or the effect of infection by *E. histolytica*, mainly in rural or suburban areas where the indiscriminate use of antibiotics can secondarily induce intestinal dysbiosis; therefore, interactions between *E. histolytica* and the microbiota are an interesting field of research.

## 7. Discussion and Future Perspectives

Although *E. histolytica* was discovered over 100 years ago, amoebiasis remains a major public health challenge around the world, especially in developing countries where a large segment of the population does not have easy access to standard sanitary conditions and, therefore, does not have good hygienic conditions, making strategies that minimize its transmission ineffective [4]. The only reservoir of *E. histolytica* is humans and the infection occurs through food, water, or hands contaminated with fecal matter that contains cysts. Therefore, it is necessary to develop government prevention and control programs of transmissibility, morbidity, and mortality caused by *E. histolytica* in endemic countries, which is a real challenge. Likewise, as mentioned in this review, person-to-person transmission also occurs through oral–genital and oral–anal contact, especially among the homosexual population, so it is necessary to implement informational strategies to avoid the transmissibility of the different forms of amoebiasis in this population.

Given the different ways amoebiasis can occur, it is important to have robust tests to help make an accurate diagnosis and overcome the limitations of the current diagnosis, especially in endemic areas, where individuals are constantly exposed to *E. histolytica*.

Advances in molecular diagnostic methodologies have been of great importance to recognize and differentiate *E. histolytica* from nonpathogenic *Entamoeba* species, and this differentiation has had a great impact on the epidemiology and clinical management of patients. However, although molecular detection is highly sensitive and specific and has the potential to be developed into field-applicable tests, the cost is still a barrier to their extensive usage as a routine laboratory test method in most endemic areas, where a correct diagnosis of amoebic infection is important to avoid morbidity, death, and disease transmission. As well, new strategies must include improving the potency of existing amebicides, discovering the new usage of approved drugs, drug rediscovery, and drug targeting of essential *E. histolytica* components to use in deworming campaigns, especially in endemic countries. Further research areas that need to be explored include the discovery of biomarkers that can predict individuals prone to symptomatic infection, distinguish between the different stages of infection, and monitor treatment success. In the same way, efforts should also be directed at studies that focus on the interactions of *E. histolytica* with microorganisms in the intestinal environment. This knowledge would have a positive impact on the clinical and laboratory diagnosis of diarrheal syndrome, its treatment, and, subsequently, the implementation of more reliable control schemes. Additionally, more research will be needed to understand the relationship between the commensal microbiota and the immune response in this parasite infection. Likewise, the advance in molecular techniques could be an important tool in the study of the geographic distribution of species of *Entamoeba* and could determine the morbidity rates of different forms of amoebic infection in different geographic areas. Although the analysis of the *Entamoeba* genomes has shed light on the study of these parasites, more research is still needed to understand, in more detail, the evolutive history of these microorganisms, in particular *E. histolytica*. In conclusion, understanding the epidemiology, pathology, and molecular biology of this parasite will not only lead to the improvement of diagnostic and treatment options but also, ultimately, to the development of safe and efficacious vaccines for the control of the disease. Finally, and no less important, the improvement of water purification systems and hygiene practices can greatly decrease disease incidence; however, it requires considerable time, changes, and monetary investments.

## Figures and Tables

**Figure 1 ijms-24-11755-f001:**
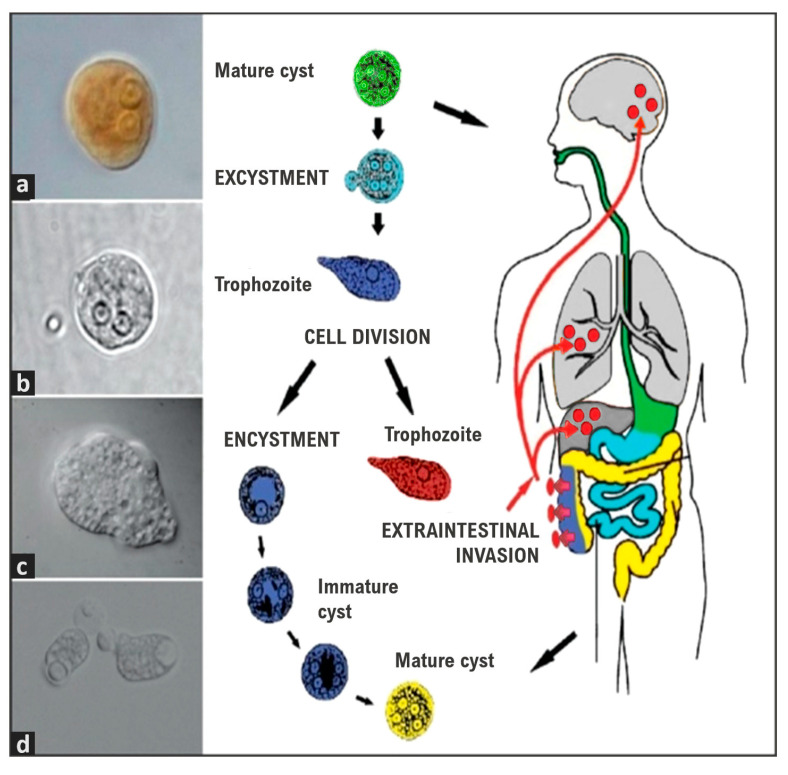
The life cycle of *E. histolytica*/*E. dispar.* Mature cysts (green) are swallowed along with food and water. The cysts pass through the stomach, but if the stomach is alkaline or when the cyst reaches the small intestine the cyst wall is damaged by trypsin, excystation starts (light blue). Trophozoites emerge (blue) and colonize the caecum and colon. Trophozoites are the infective amoeboid form which multiply by binary fission and can migrate to other tissues causing an extraintestinal invasion (red) or initiate encystment forming immature cysts and, finally, a mature cyst (yellow) to start a new cycle. (**a**) Mature cyst stained with 4% Lugol’s solution (100× magnification); (**b**) Mature cyst without staining (100×); (**c**) Trophozoite observed with differential interference contrast (DIC) (100×); (**d**) Trophozoites of *E. histolytica* with phagocyted erythrocytes (DIC) (40×) in Ximénez et al. [11].

**Figure 2 ijms-24-11755-f002:**
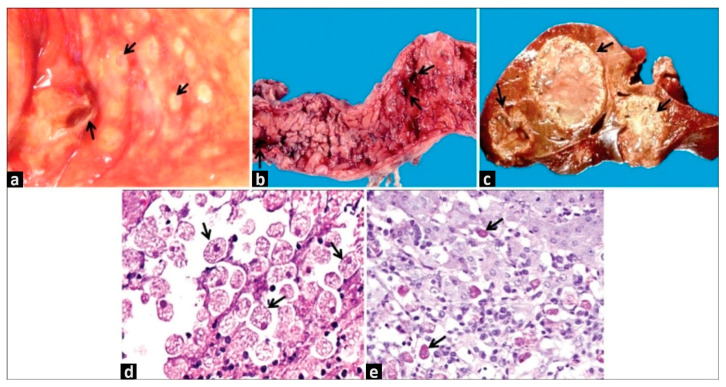
(**a**) Intestinal flask-shaped ulcers observed through rectosigmoidoscopy examination. Arrows indicate colonic ulcers. (**b**) Large bowel necropsy specimen from a case of fulminant amoebic colitis. Arrows indicate hemorrhagic ulcers and important intestinal mucosa necrosis. (**c**) Necropsy specimen of liver abscesses. Arrows indicate the three large abscesses. (**d**) Intestinal biopsy obtained from the edge of a flask-shaped ulcer where large numbers of trophozoites (HE and PAS stained, 60×) are visible. (**e**) Biopsy obtained from the edge of an amoebic liver abscess (HE and PAS stained, 20×). Notice the presence of trophozoites, hepatocytes, and many inflammatory cells. Courtesy Doctor Ruy Pérez-Tamayo in Ximénez et al. [11].

## Data Availability

Not applicable.

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
