# Peer review of "Amoebiasis: Advances in Diagnosis, Treatment, Immunology Features and the Interaction with the Intestinal Ecosystem"

_ijms, 2023, doi:10.3390/ijms241411755_

Round 1
Reviewer 1 Report
This review focuses on the current state of knowledge on the diagnosis, treatment and pathogenesis of human amoebiasis. The review presents important information on the state of the field but the organization of the review lacks clarity.
Major points:
- To avoid the duplication of information and to make the ideas flow more clearly for the reader, it would be better to rearrange the sections to the following: introduction, clinical features, diagnosis, treatments and prevention, pathogenesis, intestinal ecosystem and Entamoeba interactions, and discussion and future perspectives. This will allow the information regarding genome and protein analysis (in targets for new diagnostic techniques) just before the discussion of different virulence factors (under the current pathogenesis section)
- In the introduction, the phrase “Until today” (line 42) implies that up to this point E. histolytic was the only pathogenic strain of Entamoeba that caused amoebic invasive disease, implying that other species have now been shown to cause invasive disease. Is this the case, or is E. histolytic still the only species that causes invasive disease. Please clarify this point.
- In the pathogenesis section, (lines 146-151) do the authors mean that evidence exists in the genome implying that meiotic recombination and a sexual reproduction mechanism may exist currently in the organism or that there is evidence it occurred in the past. Please clarify this point. It would be helpful to provide an example of the evidence.
- It would be helpful to move the paragraphs describing that pathogenesis is multifactorial and the physiological events of pathogenesis (lines 183-198), earlier in the section about pathogenesis, as it provides background to how the genes highlighted (example, BspA-like proteins) contribute to virulence. Virulence is a generalized term that encompasses many different physiological functions. It would be helpful to be specific as to which physiological system is involved, if known.
- Lines154-155, it would be interesting to provide one or two examples of bacterial gene(s) that were acquired by Entamoeba via horizontal gene transfer.
- Lines 156-157, do the authors mean that genes encoding A1G1 family members were acquired by horizontal gene transfer? If so, I recommend combining this sentence with the previous one to avoid unnecessary duplication. For example, “These genes tend to be involved in metabolic processes characteristic of the anaerobic lifestyle of resident intestinal bacteria as well as genes that encode proteins in the A1G1 family, whose function is related to drug resistance in bacteria”.
- As the expression of genes from the A1G1 family is diminished in the non-pathogenic E. dispar species, is it known whether these genes are similarly diminished in the some or all of the other non-pathogenic Entamoeba species? If it is not known, then it may be useful to just mention that fact (Lines 157-160)
- It would be useful to briefly describe how genes encoding the lectin Gal/GalNAc or genes encoding amoebopores are important for virulence. What, if known, is the significance of their duplication in the genome?
- Line 200, is EhCox derived “from” prostaglandin?
- Lines 202-204, the review states that silencing the EhCox gene led to the increased expression and production of endogenous cysteine protease (CP) without altering CP gene transcripts. Is it known what causes the increased expression?
- For the increase in cytopathic activity when EhnSM1 and EhnSM2 are overexpressed, is this in individual strains overexpressing EhnSM1 or EhnSM2 or a strain overexpressing both?
- Lines 214-216, in the sentence beginning “moreover, neutral sphingomyelinases overexpressing strains were not affected on the gene expression levels….”, do the authors mean that strains overexpressing neutral sphingomyelinases do not affect the expression levels of genes for cysteine proteinase….”?
- What do ameobapore A, B, and C do during an infection? Do they create pores?
- In the paragraph about the sphingomyelinases, it would be better to discuss the effect of overexpression of EhnSM1 and EhnSM2 first and then discuss EhnSM3 instead of discussing EhnSM3 then EhnSM1 and SM2 and then EhnSM3 again.
- The paragraph (lines 220-226) on elucidating the pathogenicity from genomic studies would probably work better if it was moved earlier in the section.
- Are the virulence factors EhCP1, EhCP2, and EhCP5 unregulated in response to iron? This point is not clear. How does the upregulation compare to the positive regulation of EhCP4 and EhCP6? I was left wondering why these genes/proteins were put into separate sentences? What is different about them?
- In line 250, how does the fact that acyl-CoA synthetase is involved in the catabolism of fatty acids fit in this section about iron? Does iron affect this enzyme?
- For the section on iron, it would be helpful to make it clearer as to how all of the pathways and genes/proteins/enzymes relate to iron.
- Lines 386-433, the authors discuss ELISA in two paragraphs followed by two paragraphs on PCR but these paragraphs are separated from the earlier paragraphs describing ELISA and PCR by the paragraphs on axenic culture. It would be better to put the discussion of ELISA and PCR together.
- In Section 4.3 lines 440-441, it is stated that “there is no animal model in which the biological cycle of the parasite can be reproduced” but in the next paragraphs, animal models are described. It would be helpful to be more specific as to what is meant by reproduction of the biological cycle.
- The point of the section on non-human primates is not clear.
- For the ex vivo model of invasive intestinal amoebiasis, what aspects of the human immune response were evaluated?
- The connection with the Entamoeba that infects reptiles is not clear. What aspects of infection have been studied in that model?
- The paragraph on the advances in molecular biology allowing for a better characterization of the genome and the paragraph on the analysis of protein families would fit better with the section on pathogenesis, specifically the part describing what was found in an analysis of the genome.
- Lines 608-612, based on the increased amoebic virulence factor expression upon in vitro co-culture of E. dispar trophozoites (non-pathogenic) with S. typhimurium and the clinical results from Brazil, is the field rethinking whether other Entamoeba species, aside from E. histolytic, may play a role in colitis, under certain circumstances?
Minor points:
- In the abstract (lines 22-23), “possibility of culture trophozoites” should be changed to “possibility of culturing trophozoites”.
- In the Introduction, dyed should be died (line 53)
- Lines 55-56, the sentence as written is a little confusing. It would be clearer if split into two sentences. Example: Amoebiasis is caused by the ingestion of food or water contaminated with cysts, the infective form of E. histolytic. Following excitation, the emerging trophozoites migrate to the large intestine.
- Lines 67-69, “…Entamoeba trophozoites live as commensals, feeding on colonic microbiota and host nutrients, and forming cysts that pass-through stool to perpetuate a new life cycle by fecal-oral spreading.”
- Lines 211-212, there is a sentence fragment, “From amoebic transfectants overexpressing three of the neutral sphingomyelinase enzymes encoded in the E. histolytic genome”….
- Line 286, delete “in the first place” and lead the sentence with “the gold standard for the diagnosis of amoebiasis”.
- Lines 336-338, be careful with punctuation. “During the last 10 years, diagnostics in amoebiasis have changed dramatically, considerably improving sensitivity and specificity. Some of the current techniques are based on immunological strategies, such as ELISA, or on different modalities of polymerase chain reaction (PCR), its clear advantages….:
- Lines 353-358, be careful with punctuation and run-on sentences
- Line 369, when writing “no-pathogen strains” do the authors mean non-pathogenic strains?
- For the ex vivo model of invasive intestinal amoebiasis, what aspects of the human immune response were evaluated?
There are places where the sentence structure is confusing. Also, there are a few instances of sentence fragments and confusing punctuation.
Reviewer 2 Report
Minor reviews:
- Line 46: Correct for young adult population.
- figure 1: This figure is a bit confusing. The cycle must be well explained in the foot of the figure, as well as in the figure itself, indicate well the formation of cysts until obtaining mature cysts, and indicate the infective form, the form of dispersion...
- Line 125, 132,... there are some E. histolytica with no italics.
In general, the manuscript is well structured, but a revision of the English grammar is necessary.
Round 2
Reviewer 1 Report
The authors have answered all my questions and made appropriate changes.